# In Silico and In Vitro Analysis of Sulforaphane Anti-*Candida* Activity

**DOI:** 10.3390/antibiotics11121842

**Published:** 2022-12-19

**Authors:** Bruna L. R. Silva, Gisele Simão, Carmem D. L. Campos, Cinara R. A. V. Monteiro, Laryssa R. Bueno, Gabriel B. Ortis, Saulo J. F. Mendes, Israel Viegas Moreira, Daniele Maria-Ferreira, Eduardo M. Sousa, Flávia C. B. Vidal, Cristina de Andrade Monteiro, Valério Monteiro-Neto, Elizabeth S. Fernandes

**Affiliations:** 1Programa de Pós-Graduação em Biodiversidade e Biotecnologia da Rede BIONORTE, Universidade Federal do Maranhão, São Luís 65085-040, MA, Brazil; 2Programa de Pós-Graduação em Biotecnologia Aplicada à Saúde da Criança e do Adolescente, Faculdades Pequeno Príncipe, Av. Iguaçu No 333, Curitiba 80230-020, PR, Brazil; 3Instituto de Pesquisa Pelé Pequeno Príncipe, Av. Silva Jardim No 1632, Curitiba 80240-020, PR, Brazil; 4Programa de Pós-Graduação em Ciências da Saúde, Universidade Federal do Maranhão, Av. dos Portugueses No 1966, Cidade Universitária Dom Delgado, São Luís 65085-040, MA, Brazil; 5Universidade CEUMA, Graduation Programme, Rua Josué Montello No 1, Renascença II, São Luís 65075-120, MA, Brazil; 6Departamento de Morfologia, Universidade Federal do Maranhão, Av. dos Portugueses No 1966, Cidade Universitária Dom Delgado, São Luís 65085-040, MA, Brazil; 7Departamento Acadêmico de Biologia, Instituto Federal do Maranhão, Av. Getúlio Vargas No 2158/2159, Monte Castelo, São Luís 65030-005, MA, Brazil

**Keywords:** sulforaphane, antifungal activity, *Candida albicans*, in silico analysis

## Abstract

Oropharyngeal candidiasis/candidosis is a common and recurrent opportunistic fungal infection. Fluconazole (FLZ), one of the most used and effective antifungal agents, has been associated with a rise of resistant *Candida* species in immunocompromised patients undergoing prophylactic therapy. Sulforaphane (SFN), a compound from cruciferous vegetables, is an antimicrobial with yet controversial activities and mechanisms on fungi. Herein, the in silico and antifungal activities of SFN against *C. albicans* were investigated. In silico analyzes for the prediction of the biological activities and oral bioavailability of SFN, its possible toxicity and pharmacokinetic parameters, as well as the estimates of its gastrointestinal absorption, permeability to the blood-brain barrier and skin, and similarities to drugs, were performed by using different software. SFN in vitro anti-*Candida* activities alone and in combination with fluconazole (FLZ) were determined by the broth microdilution method and the checkerboard, biofilm and hyphae formation tests. Amongst the identified probable biological activities of SFN, nine indicated an antimicrobial potential. SFN was predicted to be highly absorbable by the gastrointestinal tract, to present good oral availability, and not to be irritant and/or hepatotoxic. SFN presented antifungal activity against *C. albicans* and prevented both biofilm and hyphae formation by this microorganism. SFN was additive/synergistic to FLZ. Overall, the data highlights the anti-*Candida* activity of SFN and its potential to be used as an adjuvant therapy to FLZ in clinical settings.

## 1. Introduction

Oropharyngeal candidiasis/candidosis (OPC) is the most common, prevalent, and recurrent opportunistic fungal infection [1] and occurs in approximately 90% of HIV+ patients [2]. Opportunistic infections can be caused by different *Candida* spp., such as *C. tropicalis*, *C. glabrata*, *C. parapsilosis*, *C. krusei* and *C. dubliniensis* [3,4]; however, *C. albicans* is described as the most frequent species associated with oral opportunistic infections [5,6]. The spectrum of *Candida* spp. infection is diverse, as these pathogens can cause from asymptomatic or mild oral disease [7] to OPC, esophagitis, vulvovaginitis, onychomycosis, cutaneous candidiasis, and even evolve into invasive or systemic diseases [8]. Although highly active antiretroviral therapy has improved the prognosis of AIDS, contributing to the decline of most opportunistic infections, HIV+ patients continue to experience significant morbidity associated with *Candida*-induced infections [9,10].

Several antifungal agents are available for OPC treatment [11]. The most frequently used azole derivatives are itraconazole and fluconazole (FLZ) (both for oral administration) due to their low cost and toxicity [12]. FLZ is considered the most effective antifungal agent in OPC [13]. However, recent studies have reported isolates of resistant *Candida* species (for review, see: [14]) in immunocompromised patients [15,16,17], and associated this event with the increased use of azoles, especially FLZ, in prophylactic therapy [18], limiting therapy success [19,20]. In this context, the search for new antifungal compounds and also therapies able to inhibit *Candida* spp. adhesion, yeast-hyphae transition, and biofilm formation, with low toxicity, are necessary and urgent.

Isothiocyanates (ITCs) are secondary metabolites generated by the hydrolysis of glucosinolates, which are found in cruciferous plants of the Brassicaceae family [21]. ITCs attract attention due to their antibacterial [22,23,24], antifungal [25], antiviral [26], anticancer [27], and chemopreventive actions [28,29] associated with moderate toxicity to normal human cells.

One of the main and promising ITCs is sulforaphane (SFN: C_6_H_11_NOS_2_), a natural phytochemical derived from cruciferous vegetables such as broccoli, cabbage, and cauliflower, with several cytoprotective effects in human cells [30,31]. SFN is anti-inflammatory [32], antioxidant [33], anti-angiogenic [34], anticancer [35,36] and neuroprotective [37,38] depending on the concentration or dose and duration of exposure. In this context, SFN benefits to human health have been investigated in different clinical trials for airway, gastrointestinal and metabolic diseases, as well as autism and cancer (for review, see: [39]).

SFN antimicrobial potential has also been investigated, especially against bacteria, with little information on its antifungal potential and mechanisms. SFN is antimicrobial against *Helicobacter pylori* [40,41] and inhibits a range of Gram (−) bacteria, including *Escherichia coli* and *Pseudomonas aeruginosa*, and Gram (+) bacteria, such as *Enterococcus faecalis*, and *Staphylococcus aureus* [42,43]. Although the antibacterial effects of SFN have been consistent across these studies, its antifungal activity is yet controversial. In fact, some studies showed that *C. albicans* is resistant to SFN [43], whilst others demonstrated that *C. albicans* and *Aspergillus niger* strains are sensitive to this compound [44,45,46]. In order to gain knowledge regarding the antifungal activity of SFN, the in silico and in vitro activities of this compound were assessed against *C. albicans* ATCC 90028 and oral isolates from HIV+ patients. In addition, the ability of SFN to potentiate the effects of FLZ against *C. albicans* was investigated.

## 2. Results

### 2.1. In Silico Analysis

#### 2.1.1. Analysis of Biological Activity

The analysis of probable biological activities indicated that SFN has a >30% (Pa > 0.3) probability of presenting 147 activities. Of these, 30 have a moderate probability (Pa > 0.5) of occurrence and eight have a high probability of occurrence (Pa > 0.7). Of the total activities identified, nine were antimicrobial (Table 1). The biological activities of FLZ were also analyzed. Of the total evaluated, 56 activities with >30% (Pa > 0.3) occurrence were identified. Of those, only eight had a high probability of occurrence (Pa > 0.7). Amongst the biological activities with >30% probability of occurrence, seven were antimicrobial (Table 1). Additional non-antimicrobial biological activities of SFN and FLZ are described in Appendix A.

#### 2.1.2. Estimated Oral Bioavailability and Expected Toxicity

To predict the oral bioavailability of SFN, its partition coefficient (water/oil; iLogP), molecular weight (MW), total polar surface area (TPSA), number of hydrogen bond donors (nHBD), and number of hydrogen bond acceptors (nHBA) values were analyzed. Our results demonstrate that SFN fits the criteria to have good estimated oral bioavailability (iLogP = 2.11; molecular weight = 177.29 g/mol; TPSA = 29.43 Å²; nHBD = 2; and nHBA = 0) (Table 2). For comparison, FLZ exhibited iLogP = 0.41; molecular weight = 306.27 g/mol; TPSA = 81.65 Å²; nHBD = 1; and nHBA = 7 (Table 2).

Table 2 also shows the expected toxic effects of SFN compared to those of FLZ. Predictive analyses suggested that SFN is not irritating or hepatotoxic but has moderate mutagenic, tumorigenic, and deleterious effects on reproduction. FLZ did not affect these parameters. The estimated LD50 was 1000 and 1410 mg/kg for SFN and FLZ, respectively (Table 2). SFN and FLZ toxicity scores were equal to 4.0, indicating they are classified as harmful if ingested at their LD50 (Table 2).

Predictive gastrointestinal absorption, permeability through the BBB, and skin permeation (log Kp in centimetres (cm)/s) are shown in Table 2. Both SFN and FLZ were suggested to be highly absorbed by the gastrointestinal tract and unable to cross the BBB. The estimated values of Log Kp were −6.38 cm/s and −7.92 cm/s for SFN and FLZ, respectively (Table 2). Both compounds were expected to be water-soluble, with Log S of −2.10 and −2.17 for SFN and FLZ, respectively. The similarity of drugs to SFN was estimated at −6.47 and 1.99 for FLZ. Drug scores were 0.25 for SFN and 0.87 for FLZ.

### 2.2. In Vitro Analysis

#### 2.2.1. In Vitro Antifungal Activity

MICs and MFCs against *C. albicans* were determined for SFN in comparison with FLZ (Table 3). MIC values for SFN were equal to 30 µg/mL for all strains (ATCC 90028, Oral 38 HIV^+^ and Oral 40 HIV^+^) except for Oral 37 HIV^+^ (60 µg/mL), whilst the MFC values ranged from 30 to 240 µg/mL (Table 3). MIC and MFC values varied for FLZ. When assessed against the ATCC strain, FLZ MIC and MFC values were 1 and 8 µg/mL, respectively (Table 3). On the other hand, the MIC values observed for the *C. albicans* clinical isolates were ≥4 µg/mL (Table 3). Also, the MFC values for FLZ were ≥64 µg/mL when assessed against these microorganisms (Table 3). The antifungal effects of SFN were also investigated in non-*Candida albicans* species (*C. parapsilosis, C. krusei*, and *C. glabrata)* in comparison to FLZ. In these fungi, SFN MIC and MFC values ranged from 1.87–60 µg/mL, whilst FLZ MIC values varied from 4–16 µg/mL (Appendix A). FLZ MFC values were >64 µg/mL for Oral 22 *C. krusei* and *C. glabrata* ATCC 2001; the other assessed non-*Candida albicans* species presented MFC values ≤64 µg/mL (Appendix A).

#### 2.2.2. In Vitro Evaluation of the Combined Effects of SFN and FLZ

ATCC 90028 and the isolates Oral 40 HIV^+^ and Oral 37 HIV^+^ were selected for further studies. A checkerboard microdilution method was used to assess the combined effects of SFN and FLZ. Their interaction was calculated and expressed as mean (Table 4) and individual FICIs (Table 5). As demonstrated in Table 4, the mean FICI values obtained indicate an indifferent effect for the combination of SFN and FLZ when assessed against *C. albicans* spp.

However, when considering the individual FICI values for each tested concentration of the compounds, we found eight additive combinations between them against *C. albicans* ATCC 90028, and seven synergistic and one additive combinations against the Oral 40 HIV^+^ isolate (Table 5). SFN exhibited the best additive effects when combined at concentrations ≤1.88 μg/mL (<MIC/16) for ATCC 90028 with FLZ at 0.5 μg/mL. On the other hand, SFN showed synergistic effects for the clinical isolate Oral 40 HIV+ at concentrations ≤7.5 μg/mL (MIC/4) with FLZ at 1.0 μg/mL. A similar effect was observed for Oral 37 HIV^+^, as the combinations between SFN at concentrations ≤15 μg/mL (MIC/4) and FLZ at MIC/4 (4 μg/mL) were synergistic.

#### 2.2.3. Effects on Hyphae Formation

Hyphae are important structures of *Candida* spp. biofilms and are essential to successful fungal colonization and invasion of the host, meaning that the greater the ability of a fungi to adhere to the host, grow hyphae and form biofilms, the higher the chances of persistent and severe infections [47,48]. Therefore, the individual and combined anti-hyphae effects of sub-inhibitory concentrations of the compounds were evaluated. As *C. albicans* oral isolates 37 HIV^+^ and 40 HIV^+^ presented similar mean and individual FICIs when combined with FLZ, the effects of the compounds on hyphae growth were investigated in *C. albicans* ATCC 90028 and oral isolate 40 HIV^+^. *C. albicans* ATCC 90028 was inhibited by SFN (MIC/2 and MIC/4) and FLZ (MIC/2) (Figure 1a–c). When assessed individually for each compound, hyphae inhibition was similar for SFN and FLZ at MIC/2 (~80%) (Figure 1c). When combined, the best effects were observed for SFN at MIC/256 *plus* FLZ at MIC/4 (73.4%). The other combinations exhibited inhibitions ranging from 52.5–70% (Figure 1c). When incubated alone with the oral isolate 40 HIV^+^, SFN (MIC/2) displayed the best inhibitory effect (100%) in comparison with FLZ (Figure 2a–c). Inhibitions of similar magnitude were observed for the following combinations: SFN MIC/256 *plus* FLZ MIC/4 and SFN MIC/64 *plus* FLZ MIC/2 (62.3%) (Figure 2c).

#### 2.2.4. Effects on Biofilm Formation

The effects of sub-inhibitory concentrations of SFN and FLZ on biofilm formation by *C. albicans* ATCC 90028 and *C. albicans* clinical isolate (Oral 40 HIV^+^) were evaluated alone and in combination at the same concentrations tested in the hyphae growth assays. Only SFN (MIC/ 2 and MIC/4) was able to impair biofilm formation by the ATCC strain (Figure 3a). The best inhibition was noted for SFN at MIC/2 (31.3%). Both FLZ and SFN per se, significantly reduced biofilm formation by the Oral 40 HIV^+^ isolate (Figure 3b). Whilst FLZ attenuated this response at both tested concentrations (MIC/2 and MIC/4; 16%), SFN only reduced biofilm formation by Oral 40 HIV^+^ at MIC/2 (13.3%) (Figure 3b).

We next assessed the combined effects of SFN (MIC/64 to MIC/264) with FLZ (MIC/2 to MIC/8) against the ATCC strain and the Oral 40 HIV^+^ isolate. Only the combined incubation of SFN MIC/128 with FLZ MIC/2 was able to attenuate biofilm formation (27.2%) by *C. albicans* ATCC 90028 (Figure 3c). On the other hand, all tested combinations decreased the ability of the Oral 40 HIV^+^ isolate to form biofilm (28.9%; Figure 3d).

#### 2.2.5. Effects on mRNA Expression of Hyphae Growth- and Biofilm Formation-Related Genes

As pronounced inhibitory effects on hyphae growth and biofilm formation were observed for SFN and FLZ (MIC/2 and MIC/4), the expression of *C. albicans* virulence genes was analyzed. Figure 4a–c demonstrates the effects of SFN or FLZ on the mRNA expression of *ASL1*, *EFG1* and *RAS1* in *C. albicans* ATCC 90028. The relative expression of all three genes to *18S* rRNA was significantly impaired by all tested concentrations of the drugs in this microorganism strain. On the other hand, whilst FLZ at MIC/4 enhanced *ASL1*, *EFG1* and *RAS1* mRNA levels in *C. albicans* Oral 40 HIV^+^, SFN-tested concentrations had no effects on the isolate (Figure 4d–f).

## 3. Discussion

FLZ is considered a low-cost drug with little toxicity [12], which is readily absorbed with high bioavailability by oral route [49]. Despite its effectiveness, recent studies have reported the isolation of resistant *Candida* species in immunocompromised patients [15,16,17] and associated this with the increased use of azoles, especially FLZ, in prophylactic therapy [18], limiting treatment success [19,20]. This surge in FLZ-resistant fungi has indicated the need for new compounds able not only to kill but also to inhibit *Candida* spp. adhesion, yeast-hyphae transition, and biofilm formation whilst displaying few toxic effects.

SFN antibacterial effects have been widely shown [40,41,42,43,46]. However, controversial data has been found in regard to its antifungal activity, with varying results on fungi sensitivity to this compound [40,41,42,43,46]. Herein, we investigated SFN in silico properties in comparison with FLZ—a first-line medication used for the treatment of *C. albicans* infections. We also determined its antifungal actions against *C. albicans,* alone and in combination with FLZ.

In silico analysis confirmed SFN antimicrobial properties, which include antiparasitic, antifungal and antibacterial actions, and mechanisms ranging from alterations of membrane composition to inhibition of fungi RNA. SFN presented an estimated DS of 0.25 with a potential moderate risk for mutagenic, tumorigenic and reproduction tract deleterious effects. These estimates are controversial, considering that SFN has been demonstrated as a cytoprotective compound able to attenuate mutagenesis and tumorigenesis and prevent abnormalities in the reproductive system [35,36,50,51,52,53]. SFN was also predicted to be lethal when ingested at doses higher than its LD_50_ (1000 mg/kg). On the contrary, although potentially harmful at doses higher than 1410 mg/kg, FLZ was not suggested to be deleterious as a tumorigenic, mutagenic, hepatotoxic or irritant agent, and neither to affect reproduction.

Prediction of skin permeation (log Kp) indicated that SFN is more likely to be absorbable by skin layers than FLZ, indicating a potential use for SFN alone or in combination with FLZ to treat skin diseases, including infections. Both compounds were considered as highly absorbed by the GI tract but not through the BBB; this later observation could be interpreted as a low chance of undesirable actions for SFN and FLZ on the central nervous system when fungal infections in this tissue are absent. However, their predicted chemical properties (iLogP, MW, TPSA, nHBD and nHBA values) [47] suggest that FLZ is less likely to penetrate the BBB. In fact, previous research indicates that SFN [54,55,56] may cross the BBB, and data obtained from studies with FLZ suggest its ability to enter the brain through the BBB; this may be enhanced in infectious diseases, which affect barrier permeability [57,58,59]. These data allow us to hypothesize that the association of low doses of FLZ and SFN may be beneficial to treat superficial mycosis, such as those affecting the skin and/or mucosa. Also, the in silico analysis corroborated previous findings showing SFN as an anti-oxidant with anti-cancer activities [33,35].

Therefore, we initially assessed the antimicrobial effects of SFN and FLZ alone or in combination. FLZ, MIC and MFC values indicated this drug is more potent than SFN against *Candida* spp. Also, eight additive combinations were observed for these compounds when tested against *C. albicans* ATCC 90028, whilst seven synergistic combinations were observed against the clinical isolate (Oral 40 HIV^+^), with the best effect seen when SFN was combined at concentrations ≤MIC/16 with FLZ at MIC/2 for the ATCC strain and ≤MIC/8 with FLZ/4 for this clinical isolate. These results confirmed SFN antimicrobial actions and showed, for the first time to our knowledge, its ability to potentiate FLZ actions when at sub-inhibitory concentrations.

Interestingly, both FLZ and SFN reduced hyphae formation by *C. albicans* at sub-inhibitory concentrations, and this effect was greatest for SFN (MIC/2). When incubated together, a greater anti-hyphae action was observed for these compounds when assessed against the ATCC strain and the clinical isolate Oral 40 HIV^+^. Indeed, hyphae growth was attenuated by more than 60%. Of note, hyphae growth is an important component of mature *Candida*-induced biofilms [60], with young hyphae already present during the intermediate phase of biofilm formation (12–30 h) [61]. In addition to a fundamental role in biofilm formation, hyphae growth is a key process in invasive infections by *C. albicans*, meaning this fungal structure is vital to fungal pathogenicity [47].

Therefore, the capacity of sub-inhibitory concentrations of SFN or FLZ to markedly halt this process is a relevant finding. Thus, we next assessed the effects of sub-inhibitory concentrations of SFN and FLZ alone or in combination on *Candida*-induced biofilm formation. SFN greatly reduced biofilm formation by *Candida* spp. at the tested concentrations, whilst FLZ was only effective against the Oral 40 HIV^+^ isolate. Similarly to that observed for hyphae growth, a more pronounced anti-biofilm action was noted when the compounds were combined.

Interestingly, FLZ at 1 µg/mL—the MIC observed for the ATCC strain and MIC/8 for the Oral 40 HIV^+^ isolate tested herein, was previously shown to prevent hyphae growth without affecting biofilm-formation by resistant *Candida* spp., following 48 h incubation [48]. Another study demonstrated that a higher concentration of FLZ (512 µg/mL) is needed to attenuate biofilm formation by *Candida* spp. [62] 48 h-post incubation. These studies differ from the observed in our study in regards to the effects of FLZ on biofilm formation following 24 h incubation. Discrepancies between the studies may be due to differences in the incubation periods with the drug. Nonetheless, we show for the first time that the combination of low concentrations of SFN with sub-inhibitory concentrations of FLZ has anti-biofilm and anti-hyphae effects on *C. albicans*. Of note, SFN was previously suggested as an inhibitor of quorum sensing in bacteria [63]—a “machinery” also important for *Candida* spp. virulence [64]. Therefore, it is possible that its antifungal activity is related to the regulation of quorum-sensing genes in fungi. To assess the ability of SFN to alter the expression of genes involved in *C. albicans* quorum sensing and virulence [65,66,67], in comparison with FLZ, the relative levels of *ALS1*, *EFG1* and *RAS1* were investigated against the ATCC 90028 strain. Both compounds inhibited the expression of all three genes when tested at MIC/2 and MIC/4, indicating that their inhibitory effects on biofilm formation and hyphae growth by *C. albicans* are associated with their ability to down-regulate these genes. Although regulation of *EGF1* by FLZ was found to be attenuated across studies, controversial data has been reported for *ALS1* and *RAS1* mRNA expressions in *Candida* spp. [68,69], suggesting the up- or down-regulation of these genes by the compound is strain-dependent. This is supported by the data presented herein, as SFN and FLZ had different actions in the expression of virulence genes when assessed against ATCC 90028 and Oral 40 HIV^+^. Nevertheless, SFN’s ability to regulate these pathways in *Candida* spp. is a novel finding which deserves attention in further studies. Also, future investigations on the effects of SFN alone and in combination with FLZ on mixed populations of microorganisms, such as those found in the oral cavity and the vagina, are worthy of being pursued.

## 4. Materials and Methods

### 4.1. In Silico Analysis

#### 4.1.1. Prediction of Biological Activities

The possible biological activities of SFN were evaluated using the online Prediction of Activity Spectra for Substances (PASS). The PASS computational tool calculates the probability of a given organic molecule being active (Pa) or inactive (Pi) on a biological target by comparing their structure to organic molecules with defined biological properties (www.way2drug.com/passonline) [70,71]. For comparison, the biological activities of FLZ were also assessed.

#### 4.1.2. Prediction of Oral Bioavailability

The SwissADME web tool (http://www.swissadme.ch/index.php# accessed on 15 September 2021) [72] was used to predict the theoretical oral bioavailability of SFN, considering Lipinski’s “Rule of Five” [73] as previously described [74]. The oral bioavailability of FLZ was evaluated for comparison.

#### 4.1.3. Estimations of Toxicity and Pharmacokinetic Characteristics

To analyze the possible toxic effects and pharmacokinetic parameters (absorption, distribution, metabolism, and excretion) of SFN, the Osiris (www.organic-chemistry.org/prog/peo/drugScore.html accessed on 15 September 2021) [70] and SwissADME (http://www.swissadme.ch/index.php# accessed on 15 September 2021) [72] web tools were used. The pharmacokinetic parameters and toxicity were predicted by comparing the chemical structures of SFN with a database containing commercially available drugs and compounds. The toxic effects were classified as mutagenic, tumorigenic, irritating, and affecting the reproductive system.

Toxic doses, given as LD50 values in mg/kg of body weight, were estimated using the ProTox web tool (http://tox.charite.de/protox_II/index.php?site=compound_input accessed on 15 September 2021) [75]. The estimated gastrointestinal absorption, permeability through the blood-brain barrier, and the skin (log Kp in centimeters (cm)/s) of SFN were evaluated by the SwissADME program (http://www.swissadme.ch/index.php# accessed on 15 September 2021) [72]. Also, the probability of SFN becoming a commercial drug (“drug score”) was calculated using the OSIRIS software (www.organic-chemistry.org/prog/peo/drugScore.html accessed on 15 September 2021).

All parameters were compared to those of FLZ.

### 4.2. In Vitro Analysis

#### 4.2.1. Microorganisms

The reference strains *C. albicans* ATCC 90028, *C. krusei* ATCC 6258 and *C. glabrata* ATCC 2001 (American Type Culture Collection) were kindly provided by the Faculdade de Odontologia de Araraquara, Universidade do Estado de São Paulo. Three *C. albicans* clinical isolates from the oral cavity of HIV+ patients (Oral 37 HIV+, Oral 38 HIV^+^, and Oral 40 HIV^+^), the isolates from the oral cavity (Oral 01 HIV^+^
*C. parapsilosis* and Oral 22 *C. krusei*), and vaginal smears of patients with vulvovaginal candidiasis (Vaginal 11 *C. glabrata* and Vaginal 14 *C. glabrata*) were donated by the microorganism collection sector of the Applied Microbiology Laboratory of Universidade CEUMA (UNICEUMA).

#### 4.2.2. Inoculum Preparation

The strains were reactivated in Sabouraud Dextrose Agar (SDA, Kasvi, Italy) for 24 h at 37 °C. The fungal inoculum was prepared in phosphate-buffered saline (PBS; Sigma–Aldrich, Gillingham, UK; pH 7.0) to achieve a cell density of 1 × 10^6^ colony forming units (CFU)/mL on a spectrophotometer (at 530 nm), equivalent to a turbidimetric scale McFarland’s score of 0.5 [76].

#### 4.2.3. Determination of Minimum Inhibitory (MIC) and Fungicidal (MFC) Concentrations

FLZ and L-SFN were purchased from Sigma–Aldrich (UK). The Minimum Inhibitory (MIC) and Fungicidal Concentrations (MFC) of SFN and FLZ were determined by the microdilution broth assay, as previously described in the document M27-A4 of the Clinical and Laboratory Standards Institute (CLSI, 2008). SFN (5000 μg/mL) was dissolved in 50% dimethylsulfoxide (DMSO; Sigma–Aldrich, UK) and then diluted in RPMI 1640 medium (Roswell Park Memorial Institute, Sigma–Aldrich, UK), containing glutamine, no bicarbonate, and buffered with sulfonic morpholino propane acid (MOPS; Sigma–Aldrich, UK) to different concentrations (0.117–60 μg/mL). FLZ was diluted in RPMI-MOPS to 0.03125–64 μg/mL. RPMI 1640 containing wells plus inoculum were used as negative control in the growth assays, and 1% DMSO (v/v) was used as vehicle control. Wells containing only RPMI 1640 medium were used for sterility control.

The fungal inoculum was prepared at 1 × 10^3^ CFU/mL in RPMI 1640 medium. Then, 100 μL of the inoculum was incubated with SFN, FLZ or vehicle for 48 h at 37 °C. After the incubation period, fungal growth was analyzed visually. MIC was defined as the lowest concentration of SFN or FLZ at which no visible growth was detected. For the determination of MFC, 10 μL of the wells with concentrations >MIC were seeded on Sabouraud Dextrose Agar (SDA; Kasvi, Italy) and incubated at 37 °C for 24 h. The fungicidal activity was defined as the one in which no growth of colonies was observed.

#### 4.2.4. Combined Effects of SFN with FLZ on Fungal Survival

The anti-*Candida* effects of SFN were evaluated against the ATCC 90028 strain and the oral isolates 37 HIV^+^ and 40 HIV^+^, in combination with FLZ, by using the checkerboard assay [77]. FLZ was tested at concentrations ranging from 0.0625 to 64 μg/mL and SFN at concentrations ranging from 0.117 to 240 μg/mL.

One hundred μL of the fungal inoculum prepared at 1 × 10^3^ CFU/mL was incubated with 50 μL of SFN and 50 μL of FLZ in microplates with different combinations of drug concentrations, at 37 °C, for 48 h. The antifungal activity was assessed as described for the determination of MICs. After incubation, SFN-FLZ interactions were determined by the Fractional Inhibitory Concentration Index (ΣFICI): FICI = (MICFLZ+SFN/MICFLZ) + (MICSFN+FLZ/MICSFN), where:

MICFLZ+SFN: MIC of FLZ when in combination with SFN;

MICFLZ: MIC of FLZ;

MICSFN+FLZ: MIC of SFN when in combination with FLZ;

MICSFN: MIC of SFN;

The FICIs were calculated for all possible combinations of different concentrations for the same isolate at which no visible growth of the microorganism was observed, and the final result was expressed as mean of the FICIs. The interaction between these drugs was classified as: Synergism: ΣFICI ≤ 0.5; additive 0.5 > Σ FICI ≤ 1; indifference 1 > Σ FICI ≤ 4.0; and Antagonism: Σ FICI > 4.0; as previously described [78].

#### 4.2.5. Biofilm Formation Assay

Biofilm formation was assessed as previously described [79]. *C. albicans* (ATCC 90028 and Oral 40 HIV^+^) were inoculated in nitrogen-based yeast medium (Yeast Nitrogen Base, YNB; Sigma–Aldrich, UK) supplemented with 50 mM glucose at 37 °C for 18 h. For this, uniform yeast cells in the exponential growth phase were centrifuged, washed twice with PBS, and adjusted to contain 1 × 10^6^ CFU/mL in PBS (0.5 of McFarland scale, at 530 nm). Aliquots of this suspension (100 μL) were transferred to a 96-well polystyrene plate and incubated for 90 min at 37 °C to allow the initial adhesion of the yeasts. After the adhesion phase, the wells were washed three times with PBS in order to remove non-adherent cells. Then, 200 μL of YNB containing 100 mM glucose and different concentrations of SFN or FLZ alone or in combination were added to the wells and incubated at 37 °C for 24 h. SFN and FLZ were individually tested at MIC/2 and MIC/4 against *C. albicans* ATCC 90028 and *C. albicans* clinical isolate (Oral 40 HIV^+^). The combined effects of SFN (MIC/64 to MIC/256) with FLZ (MIC/2 to MIC/8) against *C. albicans* were also investigated. Wells containing vehicle (1% DMSO v/v) in culture medium were used as positive controls for biofilm formation. As sterility control, YNB containing 100 mM glucose was used.

After the incubation period, the resulting biofilms were washed with PBS and collected by scraping off the bottom of each well. The biofilms were resuspended in 100 μL PBS. To quantify the yeast cells, the microdrop technique was used. For this, each biofilm sample was serially diluted, and an aliquot of each dilution (10 μL) was seeded onto SDA plates and incubated at 37 °C for 24 h. After this period, colonies were counted, and the results expressed as viable cells in CFU/mL.

#### 4.2.6. Hyphae Formation Test

The effects of SFN on hyphae formation were assessed as previously described [80,81], with modifications. *C albicans* strains (1 × 10^6^ CFU/mL; 1mL) were incubated either in RPMI 1640 medium or PBS containing 10% foetal bovine serum (FBS) supplemented with 1% DMSO (vehicle controls), FLZ or SFN (MIC/2 and MIC/4). Their combined effects were also assessed (MIC/64 to MIC/256 for SFN; and MIC/2 to MIC/8 for FLZ) against *C. albicans*. Samples were incubated under agitation (100 rpm) on glass slides placed at the bottom of each well of 24-wells plates, at 37 °C, for 48 h. After the incubation period, the slides were collected, prepared and photographed under a bright-field microscope. The number of hyphae and yeasts were counted, and the results expressed as the percentage of hyphae *per* 100 cells of *C. albicans*. Also, the percentage of inhibition of hyphae formation was calculated.

#### 4.2.7. mRNA Expression of Hyphae Growth- and Biofilm Formation-Related Genes

The effects of SFN and FLZ on the expression of *C. albicans* virulence genes (*agglutinin-like sequence 1—ALS1*; *enhanced filamentous growth protein 1*—*EFG1*; and *Ras-like protein 1*—*RAS1*) [60,61,62]. *C. albicans* ATCC 90028 or the Oral 40 HIV+ isolate (1 × 10^6^ CFU/mL; 1 mL) was submitted to the biofilm formation assay in the presence and absence of sub-inhibitory concentrations of FLZ or SFN (MIC/2 and MIC/4) for 24 h. The tested concentrations of the compounds were the same used in the biofilm formation and hyphae growth assays. Vehicle (1% DMSO; v/v)-treated wells were used as positive controls for biofilm formation.

After incubation, the cells were scraped from the bottom of each well, and the resulting suspensions, transferred to 1.5 mL tubes and centrifuged at 5000× *g*, 4 °C, for 5 min. For purification of total RNA (as previously described [82], the cell pellets were resuspended in TRIzol (Invitrogen^®^) and added of 600 µL of silica beads (0.2 mm diameter). Then, each sample was homogenized twice for 2.5 min followed by 2.5 min breaks on ice. The resulting suspensions were placed on ice and sonicated for 1 min at 15% sonication amplitude in continuous mode. The samples were centrifuged (10,000× *g*, at 4 °C, for 10 min), and the aqueous phases collected and transferred to new 1.5 mL tubes added of 200 μL of RNase-free chloroform. Samples were vortexed for 15 s, let to rest for 10 min, and centrifuged (10,000× *g*, at 4 °C, for 10 min). Three hundred µL of the supernatant of each sample were then mixed with 500 µL of RNase-free chloroform (Sigma–Aldrich, UK), vortexed for 15 s, and centrifuged (10,000× *g*, at 4 °C, for 10 min). The resulting aqueous phase was added of 500 µL of RNase-free isopropanol (Thermofisher, São Paulo, Brazil), vortexed for 15 s, and centrifuged (10,000× *g*, at 4 °C, for 10 min). The supernatants were discarded, and the pellets added of 1 mL of RNase-free 75% ethanol and centrifuged at 10,000× *g* for 5 min at room temperature. The supernatant was again discarded, and the pellets dried for 15 min for removal of residual ethanol. Samples were then resuspended in 50 µL of nuclease-free water (Thermofisher, Brazil). RNA quantities in each sample were measured by NanoDrop (Thermofisher, Brazil); samples were then kept at −80 °C to further analyse.

The total RNA (5 µg) of each sample was reverse transcribed to cDNA by using the PCR GoScript™ Reverse Transcriptase kit (Promega Corporation, USA) according to manufacturer’s instructions. The resulting cDNAs were kept at −20 °C to further analyse. One µg of cDNA was amplified by qPCR on a StepOne qPCR system (Thermofisher; Brazil; hold: 10 min at 95 °C; cycling: 40 cycles: 10 s at 95 °C, 20 s at 60 °C, and 15 s at 72 °C; melt: 68–90 °C) using SYBR Green PCR Master mix (Thermofisher; Brazil). All primers for the virulence genes *ALS1* (Reverse: ATGATTCAAAGCGTCGTTC; Forward: TTGGGTTGGTCCTTAGATGG), *EFG1* (Reverse: TTGTTGTCCTGCTGTCTGTC; Forward: TATGCCCCAGCAAACAACTG) and *Ras1* (Reverse: GTCTTTCCATTTCTAAATCAC; Forward: TATGCCCCAGCAAACAACTG), and for the housekeeping gene *18S* rRNA (Reverse: TGCAACAACTTTAATATACGC; Forward: AATTACCCAATCCCGACAC), were purchased from Integrated DNA Technologies (Iowa, USA) and used at 500 nM. Results are depicted as relative expression of each virulence gene, normalized to *18S* rRNA and calculated by the 2^−ΔΔCT^ method [83].

### 4.3. Statistical Analysis

The results are presented as the mean ± standard error of the mean (SEM). All experiments were carried out in triplicate and were obtained from three independent experiments. Significant differences among groups were determined by using one-way analysis of variance (ANOVA), followed by Bonferroni. The *p*-value < 0.05 was statistically significant.

## 5. Conclusions

Overall, the data gathered herein demonstrates that SFN is an anti-*Candida* compound once it presents antifungal and fungicidal activities. SFN also inhibits virulence factors (biofilm and hyphae growth) essential for the fungi’s ability to colonize and invade host tissues by down-regulating hyphae growth- and biofilm development-associated genes in this pathogen. Its suggested ability to cross the BBB indicates that SFN alone could be used as an alternative therapy for both deep and superficial fungal infections. Also, the marked additive/synergistic effects observed for the combination of low concentrations of SFN with sub-inhibitory concentrations of FLZ, in regards to cell survival, biofilm formation and hyphae growth, indicate this association may be an interesting approach to managing *Candida* spp. infections with possible attenuation of adverse reactions caused by these compounds.

## Figures and Tables

**Figure 1 antibiotics-11-01842-f001:**
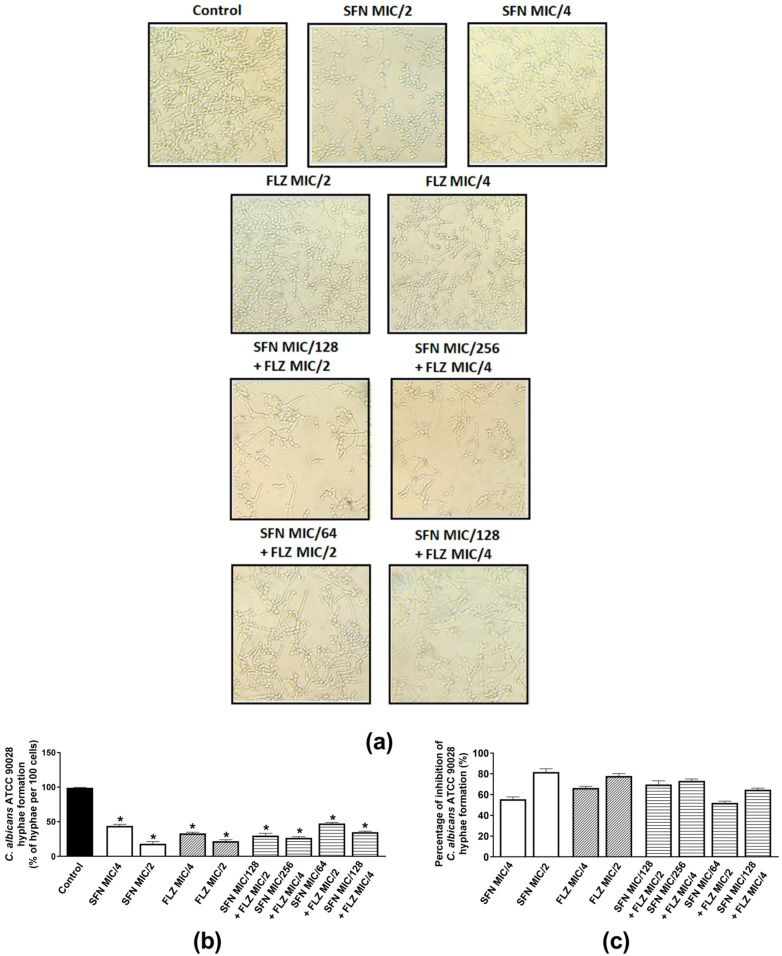
Effects of sulforaphane (SFN) and fluconazole (FLZ) on hyphae formation by *C. albicans* ATCC 90028. (**a**) Representative images of hyphae growth by *C. albicans* ATCC 90028 incubated with either FLZ or SFN at MIC/2 and MIC/4. The same panel represents the combined effects of SFN (MIC/64 to MIC/256) and FLZ (MIC/2 and MIC/4). (**b**) Percentage of hyphae *per* 100 cells of *C. albicans* ATCC 90028 and (**c**) percentage (%) of inhibition of *C. albicans* ATCC90028 hyphae formation. The control consisted of inoculum plus phosphate-buffered saline (PBS) containing 10% fetal bovine serum (FBS). * *p* < 0.05 differs from the control group.

**Figure 2 antibiotics-11-01842-f002:**
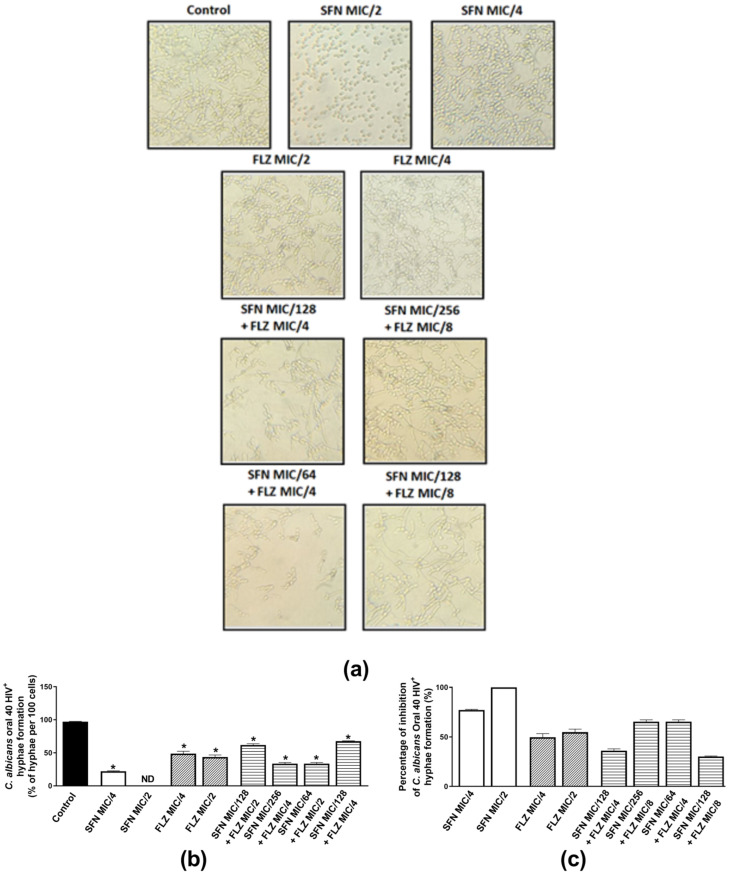
Effects of sulforaphane (SFN) and fluconazole (FLZ) on hyphae formation by *C. albicans* oral isolate 40 HIV^+^. (**a**) Representative images of hyphae growth by Oral 40 HIV^+^ incubated with either FLZ or SFN at MIC/2 and MIC/4. The same panel represents the combined effects of SFN (MIC/64 to MIC/256) and FLZ (MIC/4 and MIC/8). (**b**) Percentage of hyphae *per* 100 cells of Oral 40 HIV^+^ and (**c**) percentage (%) of inhibition of Oral 40 HIV^+^ hyphae formation. The control consisted of inoculum plus phosphate-buffered saline (PBS) containing 10% fetal bovine serum (FBS). * *p* < 0.05 differs from the control group.

**Figure 3 antibiotics-11-01842-f003:**
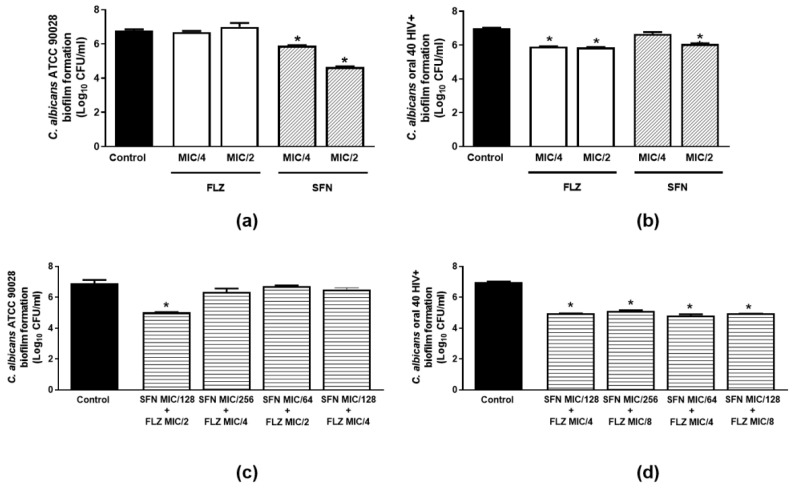
Effects of sulforaphane (SFN) and fluconazole (FLZ) on biofilm formation by *C. albicans* spp. SFN and FLZ were tested at MIC/2 and MIC/4 against (**a**) *C. albicans* ATCC 90028 and (**b**) *C. albicans* clinical isolate (Oral 40 HIV^+^). The combined effects of SFN (MIC/64 to MIC/256) with FLZ (MIC/2–MIC/8) against (**c**) *C. albicans* ATCC 90028 and (**d**) *C. albicans* clinical isolate (Oral 40 HIV+) were investigated over 24 h. The control consisted of inoculum plus culture medium. Data were obtained from three independent experiments and are expressed as Log_10_ CFU/mL.* *p* < 0.05 differs from the control group.

**Figure 4 antibiotics-11-01842-f004:**
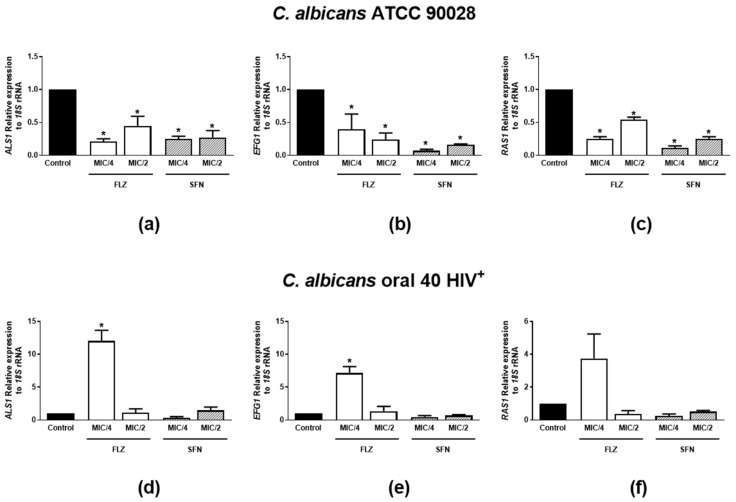
Effects of sulforaphane (SFN) and fluconazole (FLZ) on mRNA expression of hyphae growth- and biofilm formation-related genes by *C. albicans* spp. SFN and FLZ were tested at MIC/2 and MIC/4 against *C. albicans* ATCC 90028 and the oral isolate 40 HIV^+^. The relative expressions of *ALS1* (**a**,**d**), *EFG1* (**b**,**e**) and *RAS1* (**c**,**f**) normalized to *18S* rRNA were quantified 24 h following incubation with either FLZ or SFN. The control consisted of inoculum plus culture medium. Data were obtained from three independent experiments and are depicted as relative expression of mRNA. * *p* < 0.05 differs from the control group.

**Table 1 antibiotics-11-01842-t001:** In silico identification of the antimicrobial activities of sulforaphane (SFN) and fluconazole (FLZ).

SFN			FLZ		
Antimicrobial Activities	Pa	Pi	Antimicrobial Activities	Pa	Pi
Anti-*Helicobacter pylori*	0.742	0.002	Lanosterol 14 alpha demethylase inhibitor	0.846	0.001
Yeast RNA Inhibitor	0.444	0.019	Steroid synthesis inhibitor	0.744	0.001
Glycoprotein-phosphatidylinositol inhibitor	0.516	0.093	Antifungal	0.726	0.008
Antiparasitic	0.441	0.023	Phospholipid translation ATPase Inhibitor	0.480	0.069
Omptin inhibitor	0.469	0.091	NADPH-cytochrome-c2 reductase inhibitor	0.366	0.134
Phospholipid translation ATPase Inhibitor	0.363	0.140	Sugar-phosphatase inhibitor	0.356	0.148
Endopeptidase So inhibitor	0.327	0.101	Cell wall synthesis inhibitor	0.351	0.002
Mannose isomerase inhibitor	0.302	0.071			
*P. gingivalis* TPR protease inhibitor	0.301	0.108			

Pa: probability of a compound being active; Pi: probability of a compound being inactive.

**Table 2 antibiotics-11-01842-t002:** In silico estimations of the oral bioavailability, toxic effects, absorption, solubility, and drug-likeness score of sulforaphane in comparison with fluconazole.

Estimated Oral Bioavailability	SFN	FLZ
iLogP	2.11	0.41
MW (g/mol)	177.29	306.27
TPSA	29.43 Å²	81.65 Å²
nHBD	2	1
nHBA	0	7
**Predicted toxic effects**		
Mutagenic effects	Moderate	No
Tumorigenic effects	Moderate	No
Irritant effects	None	No
Hepatotoxicity	None	No
Effects on reproduction	Moderate	No
LD_50_	1000 mg/kg	1410 mg/kg
Toxicity class	4	4
**Estimated absorption**		
GI absorption	High	High
BBB permeability	No	No
Log K_p_	−6.38 cm/s	−7.92 cm/s
**Predicted solubility and drug-likeness and score**		
Log S	−2.10	−2.17
DL	−6.47	1.99
DS	0.25	0.87

BBB: blood-brain barrier; DL: drug-likeness; DS: drug-score; GI: gastrointestinal absorption; iLogP: partition coefficient water: oil–lipophilicity index; LD_50_: lethal dose 50%; Log Kp: skin permeation index; Log S: solubility; MW: molecular weight; nHBA: number of hydrogen bond acceptors; nHBD: number of hydrogen bond donors; TPSA: total polar surface area.

**Table 3 antibiotics-11-01842-t003:** Minimum Inhibitory Concentration (MIC; µg/mL) and Minimum Fungicidal Concentration (MFC; µg/mL) values of sulforaphane (SFN) compared to fluconazole (FLZ) against *C. albicans* ATCC 90028 and *C. albicans* clinical isolates (Oral 37 HIV^+^, Oral 38 HIV^+^, and Oral 40 HIV^+^).

Strain	MIC (µg/mL)		MFC (µg/mL)	
	SFN	FLZ	SFN	FLZ
ATCC 90028	30	1	60	8
Oral 37 HIV^+^	60	16	240	>64
Oral 38 HIV^+^	30	4	30	64
Oral 40 HIV^+^	30	8	60	64

**Table 4 antibiotics-11-01842-t004:** Mean fractional inhibitory concentration indexes (FICIs) for all tested combinations between sulforaphane (SFN) and fluconazole (FLZ) against *C. albicans* ATCC 90028 and *C. albicans* clinical isolates (Oral 40 HIV^+^ and Oral 37 HIV^+^).

Strains	Mean FICI (μg/mL)	Interaction
*C. albicans* ATCC 90028	2.197	Indifferent
*C. albicans* Oral 37 HIV^+^	1.412	Indifferent
*C. albicans* Oral 40 HIV^+^	1.359	Indifferent

Synergism: ΣFICI ≤ 0.5; additive 0.5 > Σ FICI ≤ 1; indifference 1 > Σ FICI ≤ 4.0; and Antagonism: Σ FICI > 4.0.

**Table 5 antibiotics-11-01842-t005:** Individual fractional inhibitory concentration indexes (FICIs) for different sub-inhibitory concentrations of sulforaphane (SFN; 0.12–30 μg/mL; MIC/256-MIC/2) and fluconazole (FLZ; MIC/2: 0.5 μg/mL for ATCC 90028, 1 μg/mL for Oral 40 HIV^+^, and 4 μg/mL for Oral 37 HIV^+^).

Strain			FIC (μg/mL) at an SFN Concentration (μg/mL) of:						
	FLZ (μg/mL)	MIC/ 256	MIC/ 128	MIC/64	MIC/ 32	MIC/16	MIC/8	MIC/4	MIC/2
ATCC 90028	**MIC/2**	0.504	0.508	0.515	0.531	0.562	0.625	0.75	1
Oral 37 HIV^+^	**MIC/4**	-	-	0.265	0.281	0.312	0.375	0.5	0.75
Oral 40 HIV^+^	**MIC/4**	0.253	0.25	0.265	0.281	0.312	0.375	0.5	0.75

Synergism: ΣFICI ≤ 0.5; additive 0.5 > Σ FICI ≤ 1; indifference 1 > Σ FICI ≤ 4.0; and Antagonism: Σ FICI > 4.0.

## Data Availability

Data will be available upon request to B.L.R.S., C.D.L.C. and S.J.F.M.

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
