# Peer review of "In Silico and In Vitro Analysis of Sulforaphane Anti-*Candida* Activity"

_antibiotics, 2022, doi:10.3390/antibiotics11121842_

Round 1

Reviewer 1 Report

The images for biofilm inhibition should be given. The current  study shows the activity of SFN only against mono-species.  Since the clinical isolates are isolated from oral samples, the activity of SFN should be proved in a simulated oral condition against mixed species of Candida and against inter-kingdom species. Gene expression studies are done only against the standard strain. It should be done against clinical strains as mixed and interkingdom species. The authours should add a note on the application of  SFN.

Author Response

# Reviewer 1

General comments

1) The images for biofilm inhibition should be given.

In order to assess the effects of SFN and FLZ on biofilm formation, following the incubation period, the resulting biofilms were collected, resuspended in PBS and the yeast cells quantified by the microdrop technique. For this, each biofilm sample was serial diluted and an aliquot of each dilution was seeded onto SDA plates. After incubation, the colonies were counted and the results expressed as viable cells in CFU/mL as previously described (Atiencia-Carrera et al., 2022; doi: 10.3389/fcimb.2022.953168). It is important to highlight that this is a well accepted method used to determine the number of viable cells composing biofilms.

2) The current study shows the activity of SFN only against mono-species. Since the clinical isolates are isolated from oral samples, the activity of SFN should be proved in a simulated oral condition against mixed species of Candida and against inter-kingdom species.

We have done our best to provide more data on SFN anti-Candida activities within the time-frame given for replying the reviewer´s comments. We performed novel experiments and included MIC and MFC data from other Candida sp. ATCC strains and isolates including C. parapsilosis and C. krusei (Supplementary Table 1). Unfortunately, we were not able to perform experiments with mixed species of microorganisms. However, we have discussed this aspect as venues to be explored by further studies in the Discussion section.

3) Gene expression studies are done only against the standard strain. It should be done against clinical strains as mixed and interkingdom species.

Within the time-frame given to reply the reviewer´s comments, we were able to perform gene expression studies with one oral isolate. The obtained results have been included and discussed in the new version of the MS.

4) The authours should add a note on the application of  SFN.

We thank the reviewer for this comment. Please note that we have now included information on SFN applications in clinical trials in the Introduction section.

Reviewer 2 Report

Please provide a graphical abstract for the article, to increase appeal and accessibility.

Please insert the chemical structures of the compounds discussed. It is very hard to discuss issues like lipophily, barrier crossing, hydrogen bond formation etc. without having the molecular structure.

Some references are rather old. Only provide references from the last 10 years.

Author Response

# Reviewer 2

General comments

1) Please provide a graphical abstract for the article, to increase appeal and accessibility.

We thank the reviewer for this comment. Please note that we have now included a graphical abstract in the new version of the MS.

2) Please insert the chemical structures of the compounds discussed. It is very hard to discuss issues like lipophily, barrier crossing, hydrogen bond formation etc. without having the molecular structure.

We thank the reviewer for this comment. We have included SFN and FLZ structures in the graphical abstract.

3) Some references are rather old. Only provide references from the last 10 years.

We agree with the reviewer on that recent references shall be included in the MS. We have replaced references 4, 5, 7, 9 and 10 by more recent literature. However, we have opted to maintain others older than 10 years, as they are important for contextualization and discussion of the data.

Reviewer 3 Report

Overall appreciation

Strengths: This study focuses on a very relevant condition as Candida infection is a problem with great impact, particularly in HIV+ patients

The methods used are adequate for the intended outputs

The use of in silico methods is very interesting and really strengthens the study

To be improved: Despite the relevance of the topic, the number of used strains was quite small. Moreover, the authors chose to work only with C. albicans. Despite being the most prevalent, non-albicans species are emerging and increasing in resistant to classic antifungals

Additionally, as the authors included a small number of strains for begin with, pursuing biofilm assays, checkerboard, and hyphal formation with only one ATCC and 1 clinical strain really limits the extrapolation of the results. At least the 4 strains should have been included, as this number was initially low.

Abstract

Overall, the abstract lacks specific results. Please include specific values obtained in the results section.

SFN was additive/synergistic – additive and synergistic are two different combinatory responses. Please clarify in the abstract for which strains was each result obtained.

into the clinics – please correct to in clinical settings.

Introduction

Sometimes it appears sp. and other times spp. Please uniformize along the text

(for review see: [14] – Please close brackets

Line 73 – the authors introduce some virulence factors associated with infection as adhesion, yeast hyphae transition, and biofilm formation. Please include a small paragraph contextualizing the importance of this virulence traits in disease severity.

(SFN: C6H11NOS2) – please correct the chemical formula

Methods

Overall appreciation: the number of included strains is small. Still, the methods are the approved standards and the complement with the in silico analyses is very interesting. Still, as the number of included strains was small, the authors must have included all four strains in the in vitro experiments.

4.2.4. Combined effects of SFN with FLZ on fungal survival

the authors tested only one isolate. Why was this isolate chosen in detriment of the others? Please clarify

Which combinations were considered to calculate the mean FIC? Only combinations below the MICs? If all combinations were included, also considering the ones without activity from both compounds, the mean will be negatively influenced? Please clarify the range of values included for the calculation of the mean FIC.

4.2.5. Biofilm formation assay

The authors choose specific combinatory concentrations to test on biofilm formation. Why decide on these combinations?

The time set of 90m to allow biofilm formation is quite small. Did the authors consider using other time endpoints to increase the time for biofilm formation? Or the authors were only considering the first step of biofilm formation (biofilm adhesion versus the effect upon preformed biofilms)?

4.2.6. Hyphae formation test

Please include the method used to analyze the hyphal formation (cell count, etc…)?

4.2.7. mRNA expression of hyphae growth- and biofilm formation-related genes

Please revise the first phrase for clarity

The concentrations (combinations) used in this assay were different from the previous assays. Please clarify the choice of concentrations.

Results

Despite the results are out of the scope of this paper, if the authors include the overall results for the prediction of biological activities, it would be interesting to include what 8 activities were predicted with a p.a.>0.7

Please consider rearranging table 3, by grouping MIC and MFC values for SFN in the first column and for FLZ in the second column.

ATCC 90028 and Oral 40 HIV+ were selected for further studies. Based on the results obtained in susceptibility testing, it would be interesting to choose the strain with a more resistant pattern to both drugs, specifically Oral 37 HIV+. Why did the authors decide on Oral 40 HIV+?

Additionally, despite the visual MIC was determined, MIC50 is the recommended breakpoint in CLSI guidelines and EUCAST guidelines. This endpoint should have been included.

2.2.3. Effects on hyphae formation

In Figure 1b. The control presents approximately 200 hyphae per 100 cells. This means that more than one hyphal structure was counted per cell. Is this correct? More importantly, the authors use concentrations that are as high as half mic. How do the authors make sure that the effect is not influenced by a reduction in growth at half mic?

The results for half MIC SFN were quite remarkable!

Discussion

The authors refer to the potential mutagenic effects of the drug. This can present a disadvantage in pursuing with SFN in future works. Did the authors find any references regarding the studies of mutagenicity of SFN or simitars? These results should be discussed and validated with experiments, at least a future perspective.

Author Response

# Reviewer 3

General comments

1) Strengths: This study focuses on a very relevant condition as Candida infection is a problem with great impact, particularly in HIV+ patients. The methods used are adequate for the intended outputs. The use of in silico methods is very interesting and really strengthens the study.

 We thank the reviewer for this comment.

 2) To be improved: Despite the relevance of the topic, the number of used strains was quite small. Moreover, the authors chose to work only with C. albicans. Despite being the most prevalent, non-albicans species are emerging and increasing in resistant to classic antifungals.

 We have done our best to provide more data on SFN anti-Candida activities within the time-frame given for replying the reviewer´s comments. We performed novel experiments and included MIC and MFC data from other Candida spp. ATCC strains and oral isolates including C. parapsilosis and C. krusei (Supplementary Table 1). We have also provided MIC and MFC data from C. glabrata (Supplementary Table 1).

 3) Additionally, as the authors included a small number of strains for begin with, pursuing biofilm assays, checkerboard, and hyphal formation with only one ATCC and 1 clinical strain really limits the extrapolation of the results. At least the 4 strains should have been included, as this number was initially low.

In the time-frame given to reply the reviewer´s comments, we could only perform additional experiments with one more C. albicans oral isolate; thus, Tables 4 and 5 contain data from the ATCC strain and the oral isolates 40 HIV+ and 37 HIV+.

Abstract

 1) Overall, the abstract lacks specific results. Please include specific values obtained in the results section.

 Unfortunately, the abstract can only contain 200 words according with the journal instructions. Therefore, we are unable to add more information in it.

 SFN was additive/synergistic – additive and synergistic are two different combinatory responses. Please clarify in the abstract for which strains was each result obtained.

 We agree with the reviewer. However, due to limitations in the number of words, we were unable to add more details to the Abstract.

 into the clinics – please correct to in clinical settings.

We have corrected it as suggested, with no changes in the word count.

Introduction

1) Sometimes it appears sp. and other times spp. Please uniformize along the text

 We thank the reviewer for this comment. We have now corrected the wording as suggested.  

2) (for review see: [14] – Please close brackets

We thank the reviewer for this comment. We have now corrected it as suggested.  

 3) Line 73 – the authors introduce some virulence factors associated with infection as adhesion, yeast hyphae transition, and biofilm formation. Please include a small paragraph contextualizing the importance of these virulence traits in disease severity.

 We thank the reviewer for this comment. As suggested, we have added a small paragraph in section 2.2.3. “Effects on hyphae formation”

 4) (SFN: C6H11NOS2) – please correct the chemical formula

Please note we have corrected SFN formulae to C6H11NOS2

Methods

1) Overall appreciation: the number of included strains is small. Still, the methods are the approved standards and the complement with the in silico analyses is very interesting. Still, as the number of included strains was small, the authors must have included all four strains in the in vitro experiments.

We have done our best to provide more data on SFN anti-Candida activities within the time-frame given for replying the reviewer´s comments. We performed novel experiments and included MIC and MFC data from other Candida spp. ATCC strains and oral isolates including C. parapsilosis and C. krusei (Supplementary Table 1). We have also provided MIC and MFC data from C. glabrata (Supplementary Table 1).

2) 4.2.4. Combined effects of SFN with FLZ on fungal survival

  The authors tested only one isolate. Why was this isolate chosen in detriment of the others? Please clarify

 Oral 40 HIV+ presented a MIC value (30 µg/mL) similar to that of most of the tested ATCC strains and clinical isolates, including non-Candida sp. For comparison, Oral 37 HIV+, found to be the most resistant (to both compounds) C. albicans tested in our study, was also investigated.  

  Which combinations were considered to calculate the mean FIC? Only combinations below the MICs? If all combinations were included, also considering the ones without activity from both compounds, the mean will be negatively influenced? Please clarify the range of values included for the calculation of the mean FIC.

 To calculate the mean FICI, the individual FICI values observed for the combinations in which there was no visible growth of C. albicans, were considered. This has been clarified in the methods.

3) 4.2.5. Biofilm formation assay

 The authors choose specific combinatory concentrations to test on biofilm formation. Why decide on these combinations?

 The effects of sub-inhibitory concentrations of SFN and FLZ on biofilm formation by C. albicans ATCC 90028 and C. albicans clinical isolate (oral 40 HIV+) were evaluated alone and in combination at the same concentrations tested in the hyphae growth assays.

 The time set of 90m to allow biofilm formation is quite small. Did the authors consider using other time endpoints to increase the time for biofilm formation? Or the authors were only considering the first step of biofilm formation (biofilm adhesion versus the effect upon preformed biofilms)?

Please note that the time set of 90 min was used to allow the strains to adhere to the plate wells. After this period, the wells were washed with PBS to remove non-adherent cells. Then, sub-inhibitory concentrations of the tested compounds were added to the wells and incubated for 24h, in order to verify their effects on biofilm formation; i.e. inhibition of biolfim formation instead of disruption of pre-formed biofilms was analysed.

4) 4.2.6. Hyphae formation test

 Please include the method used to analyze the hyphal formation (cell count, etc…)?

 In the prior version of the MS, hyphae results were described as number of hyphae in 100 cells. We have now, re-analyzed the data according with the method described in in section 4.2.6. The results are expressed as the percentage of hyphae per 100 cells of C. albicans and also, as the percentage of inhibition of hyphae formation. Accordingly, the results were re-described. It is important to highlight that SFN MIC/2 still presents the best inhibitory effects independent on the calculation method used, and that the combinations of low sub-inhibitory concentrations of SFN and FLZ produced remarkable results in both the C. albicans ATCC 90028 strain and the oral isolate 40 HIV+.       

 5) 4.2.7. mRNA expression of hyphae growth- and biofilm formation-related genes

 Please revise the first phrase for clarity

 The concentrations (combinations) used in this assay were different from the previous assays. Please clarify the choice of concentrations.

In this assays, the tested concentrations of the compounds were the same used in the biofilm formation and hyphae growth assays. This has been made clearer in the methods.

Results

1) Despite the results are out of the scope of this paper, if the authors include the overall results for the prediction of biological activities, it would be interesting to include what 8 activities were predicted with a p.a.>0.7

As suggested, we have added other biological activities found for SFN and FLZ predicted with a p.a.>0.7 (Supplementary Table 2). We have also added to Table 1, two additional antimicrobial activities for FLZ (Lanosterol 14 alpha demethylase inhibitor and Steroid synthesis inhibitor) with high probability to occur. We apologise for not adding those in the previous version of the MS.

2) Please consider rearranging table 3, by grouping MIC and MFC values for SFN in the first column and for FLZ in the second column.

 We thank the reviewer for the suggestion, but we do believe that for better visualization of the differences between compounds in regards of MIC and MFC values, table 3 as well as supplementary Table 1, MIC and MFC values should be separated as depicted.

 3) ATCC 90028 and Oral 40 HIV+ were selected for further studies. Based on the results obtained in susceptibility testing, it would be interesting to choose the strain with a more resistant pattern to both drugs, specifically Why did the authors decide on Oral 40 HIV+?

 We thank the reviewer for the suggestion. As included in the new version of the MS, C. albicans oral isolates 37 HIV+ and 40 HIV+ presented similar mean and individual FICIs when combined with FLZ. Thus, the effects of the compounds on hyphae growth and biofilm formation were only investigated in C. albicans ATCC 90028 and oral isolate 40 HIV+.

 4) Additionally, despite the visual MIC was determined, MIC50 is the recommended breakpoint in CLSI guidelines and EUCAST guidelines. This endpoint should have been included.

MIC50 is an important measure when screening large groups of microorganisms. However, both CLSI and EUCAST guidelines have set breakingpoints for Candida spp. based on MIC values. For instance, the MB60 Performance Standards for Antifungal Susceptibility Testing of Yeasts, 2017, determined breakpoints for Candida spp. Nonetheless, these, are applied for commercial drugs rather than experimental compounds such as SFN. According with the cut-offs of the CLSI guidelines, both C. albicans ATCC 90028 and the oral isolate 40 HIV+ are resistant to FLZ, as well as some of the C. parapsilosis and C. krusei samples tested in our study (MIC values for FLZ > 8 µg/ml; MB60 Performance Standards for Antifungal Susceptibility Testing of Yeasts, CLSI, 2017; https://clsi.org/media/1895/m60ed1_sample.pdf). Therefore, we have opted as most of the literature studies on novel compounds and extracts, to focus and support all assays with the observed MIC values.  

 5) 2.2.3. Effects on hyphae formation

 In Figure 1b. The control presents approximately 200 hyphae per 100 cells. This means that more than one hyphal structure was counted per cell. Is this correct? More importantly, the authors use concentrations that are as high as half mic. How do the authors make sure that the effect is not influenced by a reduction in growth at half mic?

 The results for half MIC SFN were quite remarkable!

We thank the reviewer for this comment. In the prior version of the MS, hyphae results were described as number of hyphae in 100 cells. We have now, re-analyzed the data according with the method described in in section 4.2.6. The results are expressed as the percentage of hyphae per 100 cells of C. albicans and also, as the percentage of inhibition of hyphae formation. Accordingly, the results were re-described. It is important to highlight that SFN MIC/2 still presents the best inhibitory effects independent on the calculation method used, and that the combinations of low sub-inhibitory concentrations of SFN and FLZ produced remarkable results in both the C. albicans ATCC 90028 strain and the oral isolate 40 HIV+.       

Discussion

1) The authors refer to the potential mutagenic effects of the drug. This can present a disadvantage in pursuing with SFN in future works. Did the authors find any references regarding the studies of mutagenicity of SFN or simitars? These results should be discussed and validated with experiments, at least a future perspective.

We thank the reviewer for this comment. We have highlighted the controversies between the predictions made and literature data on SFN, discussing them in the new version of the MS. We agree on that the estimates given are controversial considering that SFN has been demonstrated as a cytoprotective compound able to attenuate mutagenesis and tumorigenesis, and to prevent abnormalities in the reproductive system.

Reviewer 4 Report

Silva et al. discussed the anti-Candida activity of Sulforaphane and its potential to be used as an adjuvant therapy to FLZ in clinics. The manuscript is well-written and the authors presented exciting results. I have a few questions and concerns for the authors.

Genetically, how different are ATCC 90028 from these oral isolates? How about other C. albicans not included in this study? Do authors expect them to show similar results under the same experimental conditions? 

Fig.1abc are all very blurry. Please provide high-quality figures. 

Fig4. What statistical test was used to determine the p-value? Why is there no error bar for control groups? Line 273, please reword “p<0.05 differs from the control group“.

Author Response

# Reviewer 4

Silva et al. discussed the anti-Candida activity of Sulforaphane and its potential to be used as an adjuvant therapy to FLZ in clinics. The manuscript is well-written and the authors presented exciting results. I have a few questions and concerns for the authors.

  General comments

1) Genetically, how different are ATCC 90028 from these oral isolates? How about other C. albicans not included in this study? Do authors expect them to show similar results under the same experimental conditions?

 We have done our best to provide more data on SFN anti-Candida activities within the time-frame given for replying the reviewer´s comments. We performed novel experiments and included MIC and MFC data from other Candida spp. ATCC strains and oral isolates including C. parapsilosis and C. krusei (Supplementary Table 1). We have also provided MIC and MFC data from C. glabrata (Supplementary Table 1).

 2) Fig.1abc are all very blurry. Please provide high-quality figures.

  We have added higher quality representative panels in the new version of the MS. Hope they are improved herein.

 3) Fig4. What statistical test was used to determine the p-value? Why is there no error bar for control groups? Line 273, please reword “p<0.05 differs from the control group“.

 We thank the reviewer for this comment. Please note mRNA expression, the results are depicted as relative expression o each virulence gene, normalized to 18S rRNA, calculated by the 2-ΔΔCT method in which the control sample is taken as a value equal to 1.0 (Livak and Schmittgen, 2011; doi: 10.1006/meth.2001.1262).  

Round 2

Reviewer 1 Report

overall revision is required